# The need for high-resolution gut microbiome characterization to design efficient strategies for sustainable aquaculture production

Shashank Gupta[1], Arturo Vera-Ponce de León[1,2], Miyako Kodama[3], Matthias Hoetzinger[1,4], Cecilie G. Clausen[3], Louisa Pless[3], Ana R. A. Verissimo[3], Bruno Stengel[5], Virginia Calabuig[5], Renate Kvingedal[6], Stanko Skugor[6], Bjørge Westereng [1], Thomas Nelson Harvey[2], Anna Nordborg[7], Stefan Bertilsson [4], Morten T. Limborg [3], Turid Mørkøre[2], Simen R. Sandve[2], Phillip B. Pope [1,2,8], Torgeir R. Hvidsten [1,9] ✉ & Sabina Leanti La Rosa [1,2,9] ✉

Microbiome-directed dietary interventions such as microbiota-directed fibers (MDFs) have a proven track record in eliciting responses in beneficial gut microbes and are increasingly being promoted as an effective strategy to improve animal production systems. Here we used initial metataxonomic data on fish gut microbiomes as well as a wealth of a priori mammalian microbiome knowledge on α-mannooligosaccharides (MOS) and β-mannan-derived MDFs to study effects of such feed supplements in Atlantic salmon (*Salmo salar*) and their impact on its gut microbiome composition and functionalities. Our multi-omic analysis revealed that the investigated MDFs (two α-mannans and an acetylated β-galactoglucomannan), at a dose of 0.2% in the diet, had negligible effects on both host gene expression, and gut microbiome structure and function under the studied conditions. While a subsequent trial using a higher (4%) dietary inclusion of β-mannan significantly shifted the gut microbiome composition, there were still no biologically relevant effects on salmon metabolism and physiology. Only a single *Burkholderia-Caballeronia-Paraburkholderia* (*BCP*) population demonstrated consistent and significant abundance shifts across both feeding trials, although with no evidence of β-mannan utilization capabilities or changes in gene transcripts for producing metabolites beneficial to the host. In light of these findings, we revisited our omics data to predict and outline previously unreported and potentially beneficial endogenous lactic acid bacteria that should be targeted with future, conceivably more suitable, MDF strategies for salmon.

Developing efficient and environmentally sustainable aquaculture production systems is essential to guarantee long-term food security, especially in light of the twofold increase in global demand for seafood expected by 2050 (www.fao.org). Identification of sustainable feed ingredients that promote fish welfare and maximize growth potential, with minimal environmental impacts, is therefore a major research focus in the aquafeed industry. One interesting type of such feed ingredients have little direct nutritional value for the animal, but rather target and shift microbial populations inhabiting the gastrointestinal tract and thereby influence the feed-microbiome-host axis in a beneficial way[1,2]. These microbiome-modulating feed ingredients

¹Faculty of Chemistry, Biotechnology and Food Science, Norwegian University of Life Sciences, Ås, Norway. ²Faculty of Biosciences, Norwegian University of Life Sciences, Ås, Norway. ³Center for Evolutionary Hologenomics, The Globe Institute, University of Copenhagen, Copenhagen, Denmark. ⁴Department of Aquatic Sciences and Assessment, Swedish University of Agricultural Sciences, Uppsala, Sweden. ⁵Cargill Food Solutions – R&D – SST, Sandnes, Norway. ⁶Cargill Aqua Nutrition, Cargill, Sandnes, Norway. ⁷Department of Biotechnology and Nanomedicine, SINTEF, Trondheim, Norway. ⁸Centre for Microbiome Research, School of Biomedical Sciences, Queensland University of Technology (QUT), Translational Research Institute, Woolloongabba, QLD, Australia. ⁹These authors contributed equally: Torgeir R. Hvidsten, Sabina Leanti La Rosa. ✉e-mail: torgeir.r.hvidsten@nmbu.no; sabina.leantilarosa@nmbu.no

are well-established in terrestrial animal production[3], but are now also gaining attention in aquafeed research[4]. In salmonid feed research, there have been positive growth effects from supplementing feed with carbohydrates, and, for Atlantic salmon, fructo-oligosaccharides and α-mannooligosaccharide (α-MOS from yeast cell wall) seems particularly promising[5,6]. In addition, in vitro studies with a salmon gut simulator[7] have demonstrated that supplementation of α-MOS leads to shift in microbial community composition, with increase in lactic acid-producing *Carnobacterium*, and enhanced production of propanoic and formic acids, both of which have been demonstrated to positively impact animal microbiome and health[8].

Another emerging and potentially powerful feed design strategy is to use microbiota-directed fibres (MDFs). These compounds have chemical structures that align with specific enzymatic capabilities of certain microbial species[3]. Examples include β-mannans that are plant-derived glycans found abundantly in human and livestock diets. Depending on their sources, β-mannans have been categorized into four subtypes; linear β-mannan, galactomannan, glucomannan and galactoglucomannan[9]. Norway spruce wood-derived galactoglucomannan, for instance, was developed as an MDF in weaning piglets and was shown to be highly selective for *Roseburia intestinalis* as well as cross-feeding *Faecalibacterium prausnitzii* populations, with directed changes in short chain fatty acid (SCFA) output towards butyrate[10–12].

Despite preliminary indications via taxonomic surveys that *Carnobacterium*, *Roseburia* and *Faecalibacterium* spp. exist in the salmon gut[13], a striking paucity of genomic information for the salmon gut microbiome[14] has so far prevented the *in-depth* evaluation of the effectiveness of MDFs such as α- and β-mannans to match enzymatic abilities inherent to endogenous gut microbes. Indeed, at the inception of the present study, the understanding of the functional potential of the salmon gut microbiome was limited to microbes belonging to two salmon gut-associated genera, namely *Lactobacillus* and *Mycoplasma*, and derived from the genomes of 19 gut-associated *Lactobacillus* isolates[15] and 11 *Mycoplasma* metagenome assembled genomes[16,17] (MAGs) that met the medium-quality level of the Minimum Information about a Metagenome Assembled Genome (MIMAG) criteria (completion ≥50%, contamination ≤5%)[18]. More recently, our group has established the first resource of metagenomes and genomes from cultured bacterial strains from the salmon gut, namely the Salmon Microbial Genome Atlas (SMGA), an assemblage of 211 closed bacterial genomes and medium/high-quality MAGs obtained through cultivation and shotgun metagenomics using long-read and short-read sequencing[19]. Importantly, this resource captures the compositional and metabolic diversity of the salmon gut ecosystem, that, based on 16S rRNA amplicon sequencing, includes members of the order *Lactobacillales* (i.e. *Lactobacillus*, *Limosilactobacillus*, etc.), *Enterobacterales* (i.e. *Photobacterium* and *Aliivibrio*) and *Pseudomonadota* (i.e. *Pseudomonas* and *Burkholderia/Paraburkholderia*), as the more prevalent bacterial genera[4,20,21], as well as less abundant genera such as *Cetobacterium*, *Sphingomonas*, *Shewanella*, *Serratia*, *Lelliota* and *Actinomycetales* (*Glutamicibacter* spp)[4,21–23]. While studies limited to the assessment of compositional changes in the salmon gut have often been biased by carry-over of microbial DNA from abundant microbial populations in the feed[24,25], application of the SMGA as a database for functional omics finally enables to clearly uncover dietary effects of feed interventions, discriminating between unaffected and metabolic activated microbial populations interacting with dietary components.

Nonetheless buoyed by the circumstantial evidence surrounding mannans as an MDF, we enthusiastically pursued this application in a series of trials with the motivation that it could offer exciting prospects for aquaculture research and innovation, potentially aiding salmon feed production to enhance industry sustainability and feed efficiency. Here, we present the first implementation of our recently established SMGA resource[19], to systematically investigate functional effects of feed supplementation on the salmon holobiont (i.e. gut microbiome-salmon system). To this aim, we applied a series of host and microbiome omic analyses with the two-pronged objectives of (i) better understanding the mechanistic link between salmon gut microbes, their metabolic functions, and host physiology[26] and (ii) evaluating the potential of mannans as MDFs in salmon aquafeed. This included 16S rRNA gene profiling, metatranscriptomics, targeted metabolomics and transcriptomics of salmon organs across feeding trials with different inclusion levels of mannans and across different life stages (see method section). Our study provides evidence that α- and β-mannan-derived MDFs have negligible effects on salmon, both at the level of gut microbiome and fish physiology. We conclude that these specific MDFs do not qualify for further research as feed ingredients in salmon aquafeeds aimed at stimulating microbes present under normal rearing conditions, that is, in the absence of a pathogenic assault or an environmental or nutritional stressor. Nevertheless, the work still showcases the power of our extensive biomolecular data collection along with our newly generated SMGA[19] to devise promising avenues for testing and using feed additives that could have beneficial microbiome-reprogramming outcomes in the salmon gut.

## Results and discussion

To address knowledge gaps concerning the feed-microbiome-host axis in salmon, a trial was designed to assess individualized responses to an industry-standard 0.2% inclusion of one acetylated β-galactoglucomannan (MN3) and two different types of α-mannans (MC1 and MC2). These MDFs had varying degrees of polymerization and substrate complexity in the form of side-chain decorations and acetylation patterns (see method section). Samples were collected at different developmental stages of the fish, and varying layers of phenotypic and omics data (16S rRNA, metagenomic, (meta)transcriptomic) was generated from the gut microbiome as well as host gut tissue (Fig. 1A). Phenotypic scoring suggested the varying diets had no effect on key performance indicators (KPIs) such as weight, length, and organ integrity (Fig. 1B, Supplementary Result 1.1: Supplementary Data 4c). A series of computational analyses were performed to investigate the correlation between alterations in the expression level of genes encoding enzymes produced by diverse microbiota members, and potential shifts in nutrient utilization or uptake within the fish gut. We detected a total of 839 bacterial genera from 44 phyla using 16S rRNA gene sequencing in a global gut analysis of all fish samples; however, no MDF-driven structural changes in the gut microbiome were observed (Fig. 1C, E), not even as the microbiome evolved over time as the fish transitioned from freshwater to saltwater (Fig. 1D). An in-depth characterization of microbial functions and community level expression recovered 117,261 microbial genes, and while their activity profiles changed, as expected, over life stages (Supplementary Result 1.3), no significant clustering was observed with respect to different MDFs (Fig. 1H). Further analysis revealed only 208 significantly differentially expressed genes (DEGs) between MDF-fed fish and the control in sampled pre-smolts (T1), smolts (T2) and post-smolts (T3) (ranging from 8 to 36 per life stage). None of the DEGs was metabolically linked with mannan (Supplementary Result 1.3: Supplementary Data 4e). Finally, to see if diet-driven changes exerted any metabolic influence on their host directly or indirectly (via microbiome activity), we performed transcriptomic analyses on the gut tissue from 48 salmon that were fed either a control diet or the three MDF diets at pre-smolts (T1), smolts (T2) and post-smolts stages (T3). As expected, we found significant differences between life stages (Supplementary Result 1.4: Supplementary Fig. 7) but did not observe any significant differences in gene expression between the MDF diets and control (Fig. 1F).

To ensure that MDF inclusion levels were high enough to facilitate a diet-driven alteration of the salmon gut microbiome, we performed a small-scale re-iteration of our original trial feeding freshwater salmon a diet supplemented with 4% acetylated galactoglucomannan (Fig. 2A), a level that had proven results in monogastric animal trials[11]. In this case, the microbiome analysis was extended to include both the hindgut and pyloric caeca, which is a critical part of the salmon digestive system[27,28]. Using a similar data generation and analysis workflow, we identified 683 bacterial genera from 36 different phyla from the hindgut and 510 bacterial genera from 33

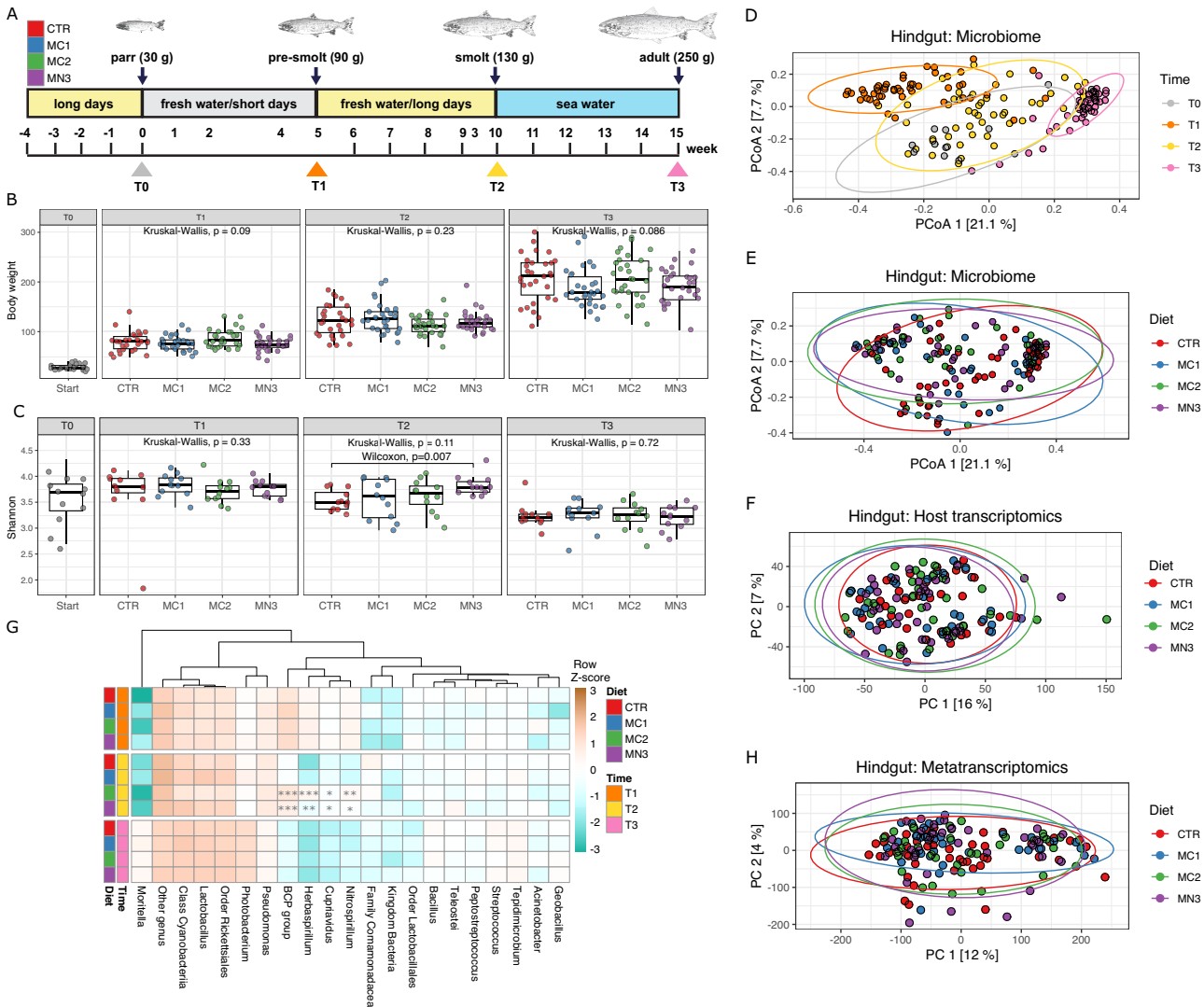

**Fig. 1 | Effect of mannan (0.2% inclusion rate) on host and gut microbial community structure and function. A** Sampling strategy for studying the effect of mannan on the temporal dynamics of the Atlantic salmon gut microbiota. T0 (parr), T1 (pre-smolts), T2 (smolts), and T3 (post-smolts) represent the different sampling time points. The experimental groups are labeled as CTR (Control), MC1 (Diet 1), MC2 (Diet 2), and MN3 (Diet 3), indicating the different diets administered to fish. **B** The mean body weight in all the experimental groups, stratified by sampling time ($n = 390$ samples). Boxplots show medians and Interquartile Range (IQR). *P*-values were determined by the Kruskal–Wallis test with False Discovery Rate (FDR) control for multiple testing. **C** Alpha diversity (Shannon diversity index), stratified by sampling time ($n = 156$ samples). Boxplots show medians and IQR. *P*-values were determined by the Kruskal–Wallis test for comparisons involving more than two groups, and the Wilcoxon test for two-group comparisons, with FDR control for multiple testing. **D** Beta diversity was assessed using Bray–Curtis dissimilarity for 16S rRNA gene data obtained from the hindgut samples ($n = 156$ samples). The effect of sampling time was tested with PERMANOVA. Each dot represents individual samples colored by sampling time (T0 (parr), T1 (pre-smolts), T2 (smolts), and T3 (post-smolts)), as indicated in the legend. **E** Beta diversity was assessed

through Bray–Curtis dissimilarity for 16S rRNA gene data obtained from the hindgut samples ($n = 156$ samples). The effect of different diets was tested with PERMANOVA. Each dot represents individual samples colored by different MDFs (CTR (Control), MC1 (Diet 1), MC2 (Diet 2), and MN3 (Diet 3), as indicated in the legend. **F** PCA plot showing the differences in host gene expression between MDFs (MC1 (Diet 1), MC2 (Diet 2), and MN3 (Diet 3)) and control samples (CTR) from the hindgut ($n = 142$ samples). **G** Top 20 most abundant genera in all the groups based on 16S rRNA gene data. Other genera with relative abundance less than 1% were shown as "Other genus". Statistically significant differences between MDFs (MC1 (Diet 1), MC2 (Diet 2), and MN3 (Diet 3)) and control samples (CTR) were calculated at each sampling point using the Wilcoxon test, with significance levels indicated by stars: *$p \leq 0.05$, **$p \leq 0.01$, ***$p \leq 0.001$. Colors represent row Z-scores of each microbial genus (brown: high; turquoise: low). A total of 144 samples were analyzed out of 156, excluding samples from T0 (parr). **H** PCA plot showing the differences in metatranscriptome expression between MDFs (MC1 (Diet 1), MC2 (Diet 2), and MN3 (Diet 3)) and control samples (CTR) from the hindgut ($n = 139$ samples). In D, E, F and H, the numbers on the axes represent the variance explained by the principal components.

phyla from the pyloric caeca samples. The 4% supplementation with galactoglucomannan (4% MN3) in the feed caused a significant change in microbiome composition (Shannon diversity; Wilcoxon, $p = 0.045$, Bray–Curtis distance metrics tested by PERMANOVA for diet, hindgut: $p = 0.013$, pyloric caeca: $p = 0.0035$) (Fig. 2C–E and Supplementary Result 1.5). Differential genus abundance analysis showed that the *Burkholderia-Caballeronia-Paraburkholderia* (*BCP*) group (Wilcoxon, $p < 0.001$) and *Pseudomonas* (Wilcoxon, hindgut: $p < 0.001$, pyloric caeca:

$p < 0.05$) taxa were significantly increased in relative abundance, while levels of *Limosilactobacillus* were reduced (Wilcoxon, $p < 0.001$) (Fig. 2F). Despite observing microbiome compositional shifts, no significant changes in host phenotype and metabolism were observed (Fig. 2B), with no significant clustering differences in gene expression profiles generated from hindgut (Fig. 2G) and pyloric caeca tissues (Supplementary Result 1.6: Supplementary Fig. 9) of fish fed either the control or experimental diet. A limited number of significant DEGs (only 6 out of 47,563 for the hindgut and 2 out

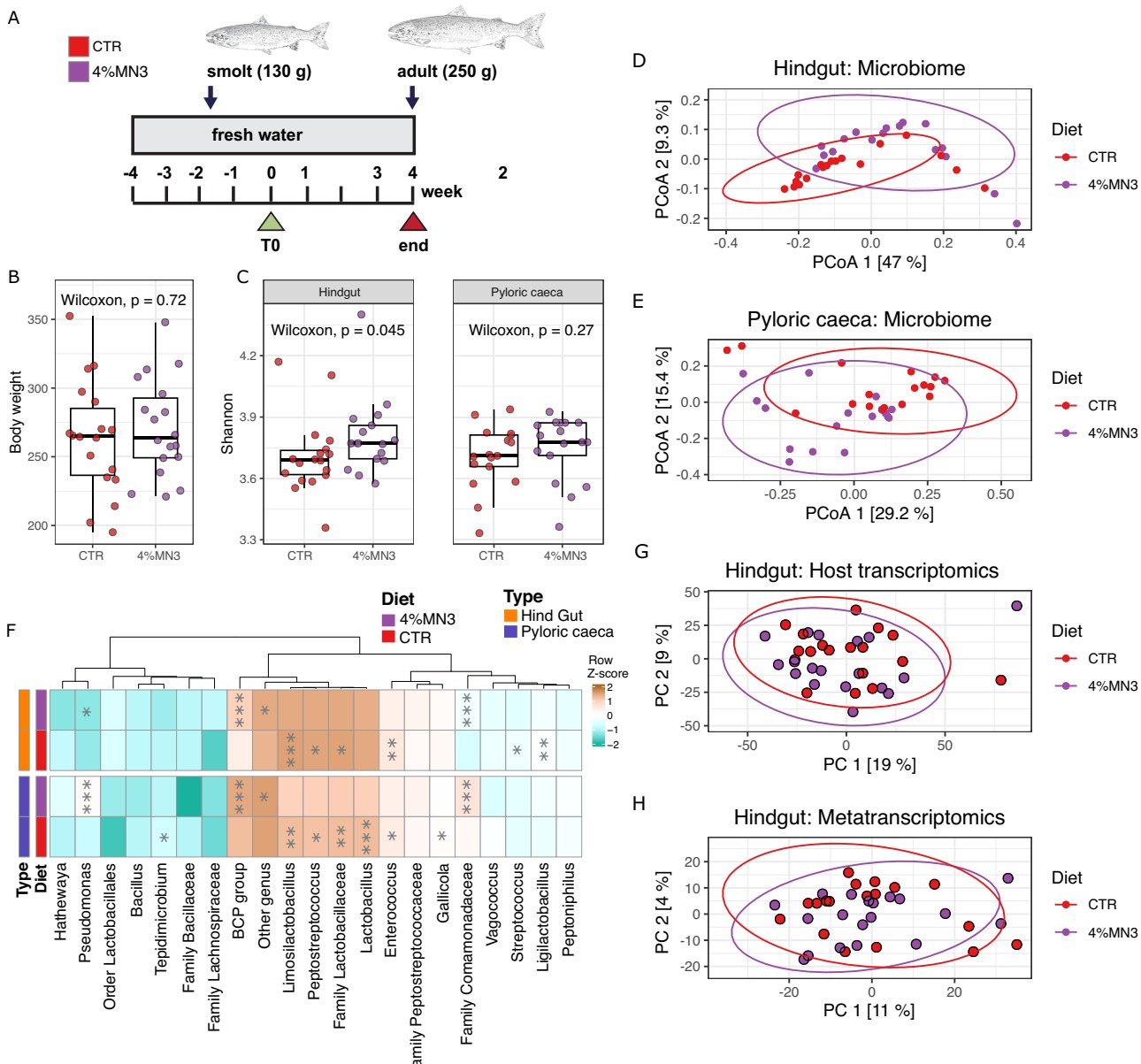

**Fig. 2 | Effect of β-mannan (4% inclusion rate) on host and gut microbial community structure and function. A** Sampling strategy for studying the effect of β-mannan on the temporal dynamics of the Atlantic salmon gut microbiota. The experimental groups are labeled as CTR (Control) and 4%MN3 diet (4% β-mannan diet). **B** Boxplot showing the mean body weight in all the experimental groups ($n = 36$). Boxplots display medians and IQR. *P*-values were determined by the Wilcoxon test with FDR control for multiple testing. **C** Alpha diversity (Shannon diversity index), stratified by sample type (hindgut, $n = 35$ samples; and pyloric caeca, $n = 34$ samples). Boxplots display medians and IQR. *P*-values were determined by the Wilcoxon test with FDR control for multiple testing. **D** Beta diversity was assessed using Bray–Curtis dissimilarity for 16S rRNA gene data obtained from the hindgut samples ($n = 35$ samples). The effects of the diet were tested with PERMANOVA. Each dot represents individual samples colored by diet, as detailed in the legend. **E** Beta diversity was assessed using Bray–Curtis dissimilarity for 16S

rRNA gene data obtained from the pyloric caeca samples ($n = 34$ samples). The effects of the diet were tested with PERMANOVA. Each dot represents individual samples colored by diet, as detailed in the legend. **F** Top 20 most abundant genera in all the groups based on 16S rRNA gene data ($n = 69$ samples). Other genera with relative abundance less than 1% were shown as "Other genus". Statistically significant differences between the 4% β-mannan (4% MN3) and control (CTR) samples were calculated using the Wilcoxon test, with significance levels indicated by stars: *$p \leq 0.05$, **$p \leq 0.01$, ***$p \leq 0.001$. Colors represent row Z-scores of each microbial genus (brown: high; turquoise: low). **G** PCA plot showing the differences in gene expression between the 4% β-mannan (4%MN3) and control samples from the hindgut ($n = 37$ samples). **H** PCA plot showing differences in metatranscriptome expression between 4% β-mannan (4%MN3) and control (CTR) samples from the hindgut ($n = 37$ samples). In D, E, G and H, the numbers on the axes indicate the variance explained by the principal components.

of 47,433 genes for pyloric caeca) were identified, none of which were linked to galactoglucomannan utilization (Supplementary Result 1.6). Metatranscriptomic analysis of the hindgut content revealed the expression of 17,094 bacterial genes in both the control group and the high dose galactoglucomannan group, with no significant clustering between them (Fig. 2H and Supplementary Result 1.7). Further analysis identified only 5 significant

DEGs, none of which were metabolically linked with mannan processing (Supplementary Result 1.7: Supplementary Data 4i). While targeted metabolomics indicated an increase in acetate concentrations in fish fed 4% MN3 (t-test, hindgut: $p = 1.39e-7$, pyloric caeca: $p = 3.216e-4$; Supplementary Result 1.8: Supplementary Data 4j), we observed no metabolic evidence that microbial fermentation pathways linked to its production

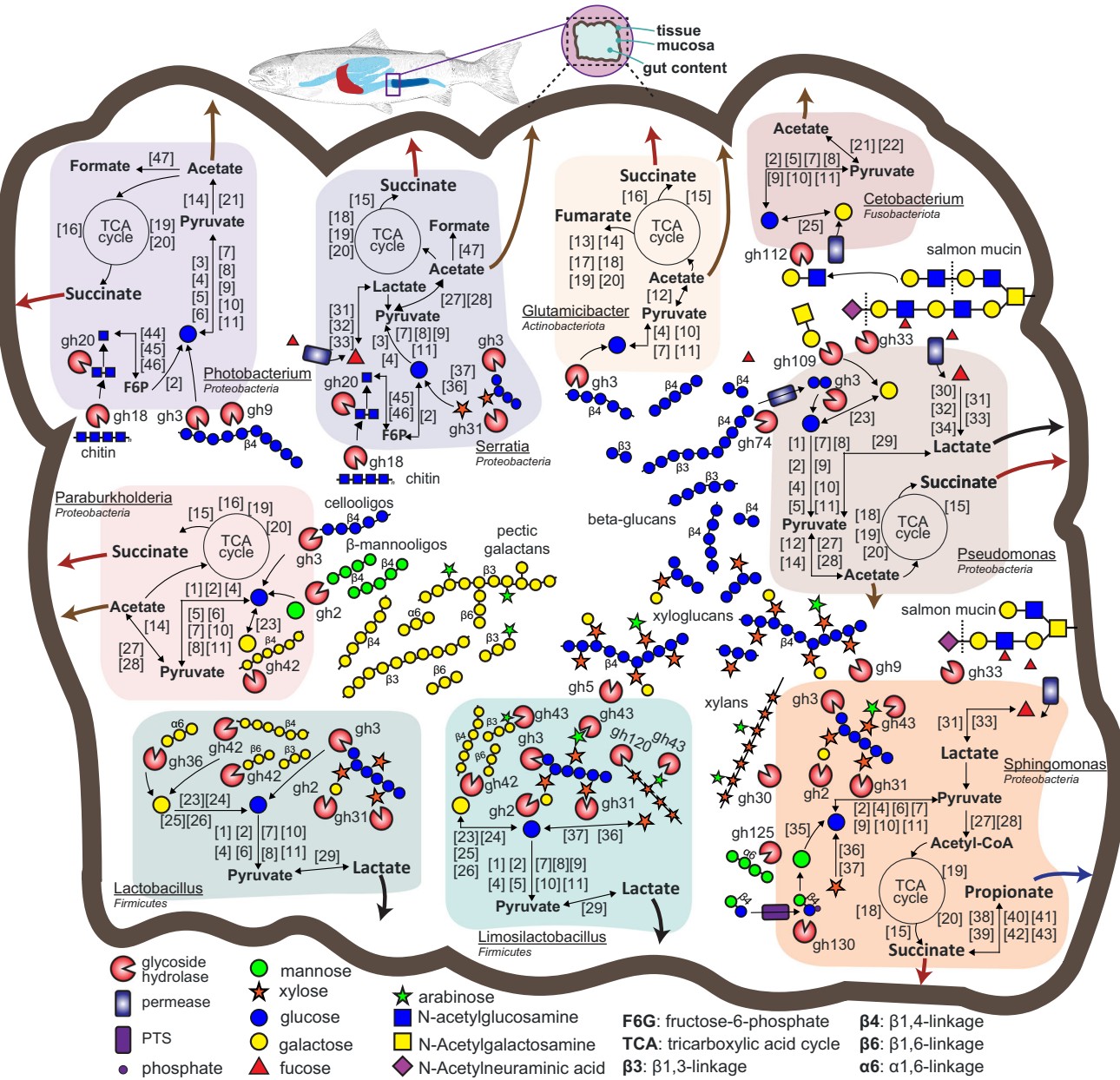

**Fig. 3 | Selected metabolic features of the salmon gut microbiome as inferred from genome and metatranscriptome comparisons.** The different metabolic pathways, including host and dietary carbohydrate depolymerization, glycolysis, tricarboxylic acid (TCA) cycle and SCFA production, are displayed for each population MAG. The graphical representation includes different carbohydrates, CAZymes, and cellular features based on functional annotations depicted as numbered boxes or abbreviated gene names, which are additionally listed in Supplementary Data 4d. Features are included if a gene was expressed at either the smolt (T2) or post-smolt (T3) stage from either the control or MDF (MC1, MC2, MN3) diets. The main carbohydrates predicted to be utilized (beta-glucans, xylans, galactans, and chitin), SCFAs (e.g., acetate), and organic acids (e.g., lactate and succinate) are represented by large colored arrows. GTBD-Tk inferred taxonomy is included. Gene names and abbreviations are also provided in Supplementary Data 4d.

were significantly increased in gene expression data, nor did we detect endo-mannanases or mannan-specific acetyl esterases (e.g. Carbohydrate Esterase belonging to family 2 and 17)[9,11,29], which theoretically could cleave acetyl decorations and depolymerize the galactoglucomannan substrate (Supplementary Result 1.5).

While we observed a subdued response of microbial and host metabolism to dietary inclusion of mannan, our multi-layered omic analyses still returned valuable information pertaining to microbial functions in the salmon gut microbiome and how they may act to benefit their host. For example, for the *BCP group* (Figs. 1G and 2F), *Limosilactobacillus* and *Lactobacillus* populations both had diet-driven changes in their abundances indicating their potential relationships to external factors that could be leveraged to facilitate their metabolic control. From the 211 microbes detected as metabolically active in the metatranscriptomics datasets (Supplementary Figs. S6 and S10), we focused on a subset of 9 randomly selected bacterial species from high abundant and less abundant genera in the salmon gut microbes, as described in publicly available studies[4,20,21,23]. Closer examination of their metabolism using genome-centric metatranscriptomics highlighted that several enzymatic features and pathways related to pectic galactans, xylans, chitin, beta-glucans, xyloglucans, host mucin-derivatives, celluloses, and undecorated manno-oligosaccharides were putatively active in the salmon gut for the low dose mannan trials (Fig. 3), with functional outputs relevant for salmon health and physiology. Moreover, salmon RNA sequencing data revealed active transcription of

transporters that could facilitate the uptake of these microbial fermentation products (Supplementary Data 3). This included the monocarboxylate transporter 1 (MCT1) SLC16A1 that facilitate uptake of lactate, propionate, and acetate into colonic cells[30]. Similarly, transporters belonging to the SLC13 family[31], that mediates succinate uptake, were also expressed, collectively supporting the notion that the host may efficiently absorbs SCFAs and organic acids produced by microbial fermentation of dietary fibers.

Among the active bacterial populations, the lactic acid-producing bacteria (LAB) *Lactobacillus* and *Limosilactobacillus*, which are renowned for their improvements to fish disease resistance via immunostimulation[32], were observed to express genes encoding xylosidases, glucosidases, and galactosidases related to xyloglucans and galactans utilization. These findings conform with their capabilities to metabolize commercially available galacto-oligosaccharides (GOS) previously identified to stimulate growth of similar LAB under in vitro culture conditions[15]. However, our in vivo detection of GH42 β-galactosidases and GH43 arabinofuranosidases points to an improved capacity of using substrates with more complex structure, such as pectin-derived galactans or (arabino)xyloligosaccharides from cereals, than many commercial undecorated GOS (synthesized from monomers obtained from animal products i.e., lactose from milk). As such, this information could be extrapolated to develop sustainable LAB-specific MDFs, by matching the chemical structure of the feed supplement to the enzymatic toolbox of these lactic-acid producing microbes (that includes enzymes able to hydrolyze carbohydrates consisting of β-1,4-linked glucose and decorated with galactose, xylose and arabinose units, or polymers consisting of β-1,3-linked or β-1,4-linked galactose with arabinose substitutions), eventually promoting their growth and beneficial outputs[33].

## Conclusions

Whilst confronted with promising preliminary 16S rRNA sequencing data suggesting that mannans would selectively target beneficial populations allegedly inherent to the salmon gut microbiome, our findings instead showed that their dietary inclusion had negligible impact on microbial functionalities and host physiology and metabolism. This was largely demonstrated by in-depth analyses of trial KPIs, microbial (meta)genomes as well as both host and microbial RNA and metabolites. Although we observed a scarcity of mannan-degrading capabilities, this study achieved an in-depth understanding for how diet and specific microbial populations interact, which should inform future MDF design and testing of their effects on the salmon holobiont. Notably, while metataxonomy data indicated an effect of the mannan diet on the *Burkholderia-Caballeronia-Paraburkholderia* population, these results did not go beyond bacterial proportions, and no empirical evidence of diet-microbiome interaction was obtained. Instead, our study revealed the active roles of beneficial microbial groups, including *Lactobacillus* and *Limosilactobacillus*, highlighting their contributions to nutrient utilization, potentially producing metabolites associated with host health states[15]. We hypothesize that trials with MDFs such as pectin-derived galactans or (arabino)xyloligosaccharides designed around the obtained microbiome information will result in more concrete insights about the metabolic roles of beneficial gut-associated microbiota and host effects in salmon and other fish in the instances they are implemented.

Overall, our experiences highlight the risk of inferring functional outcomes from 16S amplicon data and clearly reinforce the impact that critical metabolic insight can have on microbiome intervention strategies, strengthening the need for high-resolution, host-specific, microbiome characterization as a prerequisite to improved animal trial design. Further, our results also highlight the value of focusing on microbes that are naturally present in the host species of interest and the metabolic capacity of these microbes towards identifying the next pre- and probiotic candidates. While establishing a baseline understanding of the host's microbial community under normal healthy conditions, the next crucial step is to characterize changes in microbiota in altered states, such as under environmental stressors or in a disease state, with the goal to identify and broaden potential targets for intervention by MDFs. Criticisms towards metataxonomy-based microbiome characterization as being overly descriptive or merely stamp-

collecting are easy to make, but the value of microbial genome atlases and culture biobanks[19] cannot be exaggerated, especially as industry makes increasingly strong moves towards precision microbiome interventions as a viable technology to improve aquaculture sustainability and production.

## Methods

### Animal welfare

The experiments were conducted according to the guidelines and protocols approved by the European Union (EU Council 86/609; D.L. 27.01.1992, no. 116) and by the National Guidelines for Animal Care and Welfare published by the Norwegian Ministry of Education and Research. We have complied with all relevant ethical regulations for animal use.

### Experimental design and animal management

For the first feeding trial (hereafter referred to as "low dose mannan trial"), Atlantic salmon (*Salmo salar*) with an initial weight of 29 ± 0.97 g were used. The fish were randomly distributed into 12 tanks (1 m diameter and 0.45 m³ volume) supplied with freshwater (flow-through) at Cargill Innovation Center, Dirdal, Norway. The salmon were fed either a standard commercial diet (referred to as CTR diet) or the same diet supplemented with 0.2% w/w α-mannan produced either by Cargill (referred to as MC1 and MC2 diet, respectively) or spruce-wood derived acetylated galactoglucomannan produced at the Norwegian University of Life Sciences (referred to as MN3 diet). Methods for production of β-mannan from Norway spruce chips have been described in a previous study[29]. The exact structural composition of the MC1 and MC2 α-mannans is proprietary and unavailable. The inclusion level was selected based on communication with the feed industry, where a supplementation of 0.2% of MDF in feed would be realistic in a production setting for this particular ingredient. The formulation and chemical composition of the experimental diets are shown in Supplementary Data 4a.

Fish was acclimatized to experimental conditions for 28 days before the trial began. Salmon was then fed the four diets in three replicate tanks over a period of 110 days (August-December 2020), using an automatic belt feeder with continuous feeding for 20 h/day in excess of satiation level. Feed intake was calculated on a weekly basis by collecting and weighing uneaten pellets at the bottom of the tanks, using a feed collecting system, as well as by weighing the amount of feed. A 12 h light:12 h dark photoperiod regime was used from day 0 to day 33 (5 weeks) after which a 24 h light regime was used to initiate smoltification. At day 70 (10 weeks), fish from all the tanks were transferred to 12 seawater flow-through tanks and fed one of the four diets for an additional 5-week period. The salmon were sampled for analyses four times during the feeding trial; on August 8th, 2020 (T0, parr); September 24-25th 2020 (T1, pre-smolts); October 29-30th 2020 (T2, smolts) and December 7-8th 2020 (T3, post-smolts) (Fig. 1A).

The second fish trial (hereafter referred to as "high dose β-mannan trial") was conducted at the Center for Fish Research, NMBU, Ås, Norway. A total of 15 Atlantic salmon individuals (130 g of average initial weight) were placed in each of nine tanks (300 L fiberglass tanks with averaged 14.5 °C recirculated freshwater at a flow rate that keeps the oxygen level above 80% saturation). Fish was acclimatized to experimental conditions for 28 days before the trial began and fed *ad libitum* (i.e., 10% excess) with a control diet. At day 29, the fish (approx. 220–250 g average weight) were fed one of two diets: a control diet (CTR) and an experimental diet supplemented with 4% β-mannan obtained from Norway Spruce wood (4%MN3). The diets were produced by extrusion and subsequent vacuum coating with fish oil at the Centre for Feed Technology, Norwegian University of Life Sciences, Ås, Norway. The formulation and chemical composition of the two diets are shown in Supplementary Data 4b.

Each diet was tested in triplicate tanks over 28 days (23rd March-21st April 2022, 4 meals/day) using automatic feeders. No mortality or abnormal behavior in any of the fish was recorded during the experimental period.

### Sampling of fish and phenotype scoring

All fish were anaesthetised using tricaine methanesulfonate (MS-222) (80 mg/L) and bulk weighed to facilitate the calculation of growth. In the low

dose mannan trial, at each time point, ten fish were randomly selected from each tank for examination of biometric traits, fish welfare, and sampling of gut tissues and gut contents. Each fish was scored for eye opacity on a scale from 0 to 4, where score 0 represents no visible opacity and score 4 represents the opacity of more than 75% of the cross-sectional area of the lens (complete cataract)[34]. The results are presented as the sum of the opacity score of both eyes (i.e., max score is 8). The fish was thereafter placed in a light box with standardized light conditions from light emitting diodes (Led Studio Box, Model Z3030, 6000 LUX, Mickcara check), and photographed for subsequent determination of parr marks and welfare traits; scale loss, skin hemorrhages, protruding eyes, tail fin damage, tail fin hemorrhages, snout bleeding, upper jaw deformity, lower jaw deformity, shortened opercula, eye bleeding, damaged dorsal fin, damaged pelvic fin, bleedings of pelvic fins, skeletal deformities. The evaluations were largely performed according to Noble et al.[35], who defined each of the operational welfare indicators (OWI´s) as level 0 for little or no evidence of the OWI (i.e., normal), level 1 as minor, and level 3 as clear evidenced OWI. Because skin hemorrhages, protruding eyes, and hemorrhages of the tail fin, snout and pelvic fins were either absent or sparsely observed, these parameters are excluded from the results presented. The weight of whole fish, gutted fish, livers, and hearts were recorded after photographs, in addition to fish length and gender. The color of the livers was scored according to a scale from 1 to 5, where score 1 is pale/yellowish color and score 5 is dark brown. The amount of visceral fat (visible fat, VF) was similarly scored from 1 to 5 according to the visibility of pyloric caeca (PC), where score 1: PC clearly visible, score 2: PC visible, score 3: PC visible as cracks in the VF, score 4: PC visible through the VF, score 5: PC not visible[36]. VF on the heart surface was scored according to a scale from 0 to 3, where score 0: no VF, score 1 visible VF on the heart surface, score 2: severe accumulation of VF on the heart surface, and score 3: heart not visible due to severe fat accumulation. The firmness of livers and hearts was subjectively determined by an experienced researcher (score 1 for firm and score 0 for soft texture). The hindgut of each sampled fish was rinsed in phosphate-buffered saline (PBS) and carefully dried with paper. Thereafter the hindgut tissue was photographed in the previously described led light box for subsequent determination of hemorrhages of the hindgut and anus (score 0-1), general color (light/dark color; score 0, score 1) and absence of mucosal folding (i.e., macroscopic examination of hindgut appearance and architecture).

A total of 30 salmon were sampled at the first time point (T0), followed by the sampling of 120 salmon individuals (30 for each diet) at each subsequent time point (T1 – 5 weeks after the trial start; T2 – 10 weeks after the trial start; T3 – 15 weeks after the trial start). The distal gut section was dissected using sterile scalpels and tweezers, and approximately 100 mg of gut content was collected from each fish using inoculation loops. In total, 390 samples were collected for microbial community profiling, which was then preserved and stored in DNA/RNA SHIELD (Zymo Research, Irvine, CA, USA) at −80 °C until processing.

At the end of the high-dose β-mannan trial, six fish from each tank were randomly sampled. Gut content collection from pyloric caeca and distal gut sections was performed as described above.

## DNA extraction and 16S rRNA gene sequencing

Feeds and salmon gut contents for both the low and high dose mannan trials were profiled for bacterial community composition using a custom workflow developed by DNASense AP (Denmark). DNA was extracted using the methods reported in ref. 20. Metabarcoding was carried out by amplifying the V4 region of the bacterial 16S rRNA gene, using the primers 515f (GTGYCAGCMGCCGCGGTAA) and 806r (GGACTACNVGGGTWTCTAAT). Negative controls were included for extraction and PCR amplification procedures. All final PCR products were purified using the standard protocol for CleanNGS SPRI beads (CleanNA, NL) with a bead-to-sample ratio of 4:5. Concentrations were then determined using the Qubit dsDNA HS Assay kit (Thermo Fisher Scientific, USA). The purified sequencing libraries were pooled in equimolar concentrations and diluted to 2 nM. The samples were paired-end sequenced (2 × 300 bp) on a MiSeq (Illumina, USA) using a

MiSeq Reagent kit v3 (Illumina, USA) following the standard protocol, including 10% PhiX internal control. The sequencing output was generated as demultiplexed fastq-files for downstream analysis.

## Bioinformatic processing of 16S rRNA gene sequencing data

Primers were removed from the raw sequence data using "cutadapt"[37]. Further, reads were analyzed with DADA2 as implemented in the QIIME2[38] (qiime2-2021.8) pipeline to infer the amplicon sequencing variants (ASVs) present and their relative abundances across the samples. Forward and reverse reads were truncated at 280 bp and 260 bp. Other quality parameters in DADA2 were left at default values. Taxonomy was assigned using a pre-trained Naïve Bayes classifier with the Silva database (release 138, 99% ASV)[39].

Data analysis was conducted in R[40] v.4.2.2. Initial preprocessing of the ASV table was conducted using the phyloseq package[41] (v1.42.0). Further filtering was done by removing ASVs without phylum-level classification, or assigned to Archaea, Eukaryota, Mitochondria, Chloroplast. To ensure that analyses were not confounded by spurious results, we first analyzed the alpha and beta diversity of negative extraction control samples that produced sequencing reads (Supplementary Fig. 1A–C). The DNA extraction controls had significantly lower observed richness than all samples (Kruskal–Wallis test, $p = 1.7e{-}05$). Furthermore, profiles were significantly different for bacteria by different diet groups (PERMANOVA for Bray–Curtis, $p = 1e{-}04$, R2 = 0.06681). Sequencing contaminants (86 of 7521 bacterial ASVs) were identified based on the prevalence of ASVs in the negative extraction control and removed using the *decontam* package[42] (v1.18.0) with default parameters. We then removed the extraction controls before downstream analysis. To avoid biased comparison due to variable sampling depth, the ASVs table was transformed into relative abundance by dividing each value by the total number of reads in that sample. The resulting table included 6468 ASVs for the low dose mannan trial and 3505 ASVs for the high dose β-mannan trial. All downstream analyses were performed on normalized ASV tables as described in Gupta et al.[43]. Briefly, alpha diversity was estimated using the Shannon diversity index, a measure of overall microbiota richness and evenness. Furthermore, beta diversity was calculated using Bray–Curtis, weighted and unweighted UniFrac metric and visualized by principal coordinates analysis (PCoA). Alpha and beta diversity of each individual sample were determined using phyloseq[41] (v1.42.0) and visualized with ggplot2[44] (v3.4.1) in R (v4.2.2). Group differences in beta diversity was assessed using permutational multivariate analysis of variance (PERMANOVA) using the *vegan* package (v2.6.4)(https://cran.r-project.org/).

## Host transcriptomics: RNA extraction, library preparation, sequencing, and processing

For the low dose mannan trial, all samples were preserved in DNA/RNA SHIELD™, obtained by Zymo Research, following the Zymo Research standard procedure. RNA extractions were carried out at the Center for Evolutionary Hologenomics, Globe Institute, University of Copenhagen, Denmark. Hindgut total RNAs were isolated using the Zymo Research Quick-RNA Miniprep Plus Kit according to the manufacturer's instructions. For the high dose β-mannan trial, RNA extractions from pyloric caeca and hindgut samples were conducted using the RNAdvance Tissue kit (Beckman Coulter, Inc.) following the manufacturer's protocol (PN B66716AD). RNA concentration and purity were determined using a Qubit 3.0 fluorometer following the manufacturer's protocol. RNA integrity was checked by using an Agilent 2100 Bioanalyzer (Agilent Technologies, Santa Clara, CA, USA). Samples with a RIN (RNA integrity number) equal to or above 2 were used. For both trials, samples were randomized, and the library preparations were carried out by Novogene (Beijing, China) using a TruSeq Stranded mRNA kit (Illumina, San Diego, CA, USA), as per the manufacturer's protocol. Briefly, polyA-containing mRNA molecules were isolated, using magnetic oligo dT-beads, from the total RNA according to the polyA selection method and then fragmented with the fragmentation buffer. Cleaved RNA fragments were primed with random hexamers into first-

strand cDNA using reverse transcriptase and random primers. After a double-stranded cDNA was synthesized, end-repair, phosphorylation and 'A' base addition, and adapter ligation followed according to Illumina's library construction protocol. Libraries were sequenced on the Illumina NovaSeq 6000 platform at Novogene, (Beijing, China), using 300 bp paired-end sequencing. Three extraction negatives and two library negatives were included.

The sequence quality of raw RNA-Seq data was assessed using FastQC (v0.11.3). Quality trimming was performed using AdapterRemoval[45] (v2.1.3) with "trimns" and "trimqualities" options to remove stretches of mixed low-quality bases and/or Ns. Sequences shorter than 25 bp, and all unpaired reads were excluded from subsequent analyses. The quality of trimmed sequences was checked again using FastQC (v0.11.3).

Trimmed reads were aligned to the *Salmo salar* reference genome (GCF_905237065.1) using STAR[46] (v2.7.2) with two-pass alignment mode and default parameters. Aligned reads were used to generate a gene-specific count matrix across samples using the feature-Counts software[47]. Genes with 50 or more reads in total across the samples were considered for further analysis. Normalization of the counts (VST normalization) and differential expression analysis were performed using the DESeq2[48] Bioconductor R package (v1.38.3).

## Metatranscriptomics: RNA extraction, library preparation, sequencing, and processing

For the microbial RNA extraction, we followed an in-house developed protocol described in Bozzi et al.[20]. The RNA was quantified using a Qubit® RNA HS Assay Kit and a Qubit® 3.0 Fluorometer (Invitrogen™), following the manufacturer's instructions. The RNA extracts were shipped to Novogene (Cambridge, UK) for preparation of metatranscriptome strand-specific libraries, pooling, and sequencing on the Illumina NovaSeq 6000 platform to obtain 12 Gb of PE 150 bp data per sample.

The resulting sequence reads were filtered for quality using fastp (v0.12.4) with an average Phred threshold of 30 (-q 30). rRNAs and tRNAs was removed from the reads using SortMeRNA[49] (v4.3.482) with the following Silva databases[39]: silva-bac-16s-id90, silva-arc-16s-id95, silva-bac-23s-id98, silva-arc-23s-id98, silva-euk-18s-id95, silvaeuk-28s-id98 and the parameters: --out2 --paired_out –fastx. To remove all sequences derived from the fish host, the filtered reads were aligned to the *Salmo salar* reference genome (GCF_905237065.1) using the STAR[46] v 2.7.9a alignment suit. All non-mapped reads were retrieved from the sam files using Samtools[50] (v1.13) and the parameters -f 12 -F 256 -c 7. These reads were used to quantify the expression of ORFs encoded by the genomes and MAGs included in the Salmon Microbiome Genome Atlas (SMGA; that also includes 68 MAGs obtained from samples derived from the low dose mannan trial)[19] using kallisto[51] (v0.44.0). The resulting transcripts per million (TPM) abundance tables of each metatranscriptomic sample were gathered into a single table using the Bioconductor tximport 1.26.1 library in R[40] v4.2.3. A bacterial gene from the SMGA was considered expressed if it showed a value higher than one TPM in at least one replicate of the experiment. Variations in the SMGA bacterial gene expression among samples were visualized in terms of Z-scores in a heatmap generated using the pheatmap function in R.

Additionally, a de novo metatranscriptomic assembly was performed. Briefly, after fastp (v0.12.4) processing, rRNAs filtering, and removal of the host *(S. salar)* sequences, the resulting high-quality reads were assembled using Megahit[52] (v1.2.9) with the --no-mercy parameter. Assembled contigs were taxonomically classified by kraken2[53] using the standard PlusPFP database (downloaded in July 2022). Genes in bacterial contigs were functionally annotated using DRAM[54] (v1.3). A database consisting of the annotated, de novo assembled, contigs was used to quantify bacterial gene expression with kallisto[51] (v0.44.0). The resulting abundance tables for each fish sample were pooled into a single table using the Bioconductor tximport (v1.26.1), as described above. Furthermore, genes with 50 or more read-counts in total across the samples were considered for differential expression analysis. Normalization of the expressed counts (VST normalization) and differential expression analysis were performed using the DESeq2[48] Bioconductor R package (v1.38.3). This dual approach, integrating SMGA database utilization and de novo assembly, enriches the study by capturing both known and previously undetected components of the microbial community.

To create the portrayal of the metabolic features of 9 salmon gut microbes shown in Fig. 3, we randomly selected a subset of bacterial species that represent high abundant and less abundant genera in the salmon gut microbes, and that were detected as metabolically active in the genome-resolved metatranscriptomic data. For this subset, we manually parsed carbohydrate active enzymes[55], and based on the predicted functions, substrate utilization and downstream metabolism of individual monosaccharides was inferred, while calling and manually confirming that each gene involved in individual metabolic pathways was detected as differentially expressed in the metatranscriptomics data. The complete list of gene names and abbreviations depicted in Fig. 3 is provided in Supplementary Data 4d.

## Short-chain fatty acids (SCFAs) analysis

Pyloric caeca and hindgut content samples obtained from fish in the high dose β-mannan trial were subjected to targeted SCFA analysis. Sample analysis was carried out by MS-Omics (Vedbæk, Denmark). Samples were acidified using hydrochloride acid, and deuterium labeled internal standards where added. All samples were analyzed in a randomized order. Analysis was performed using a high polarity column (Zebron™ ZB-FFAP, GC Cap. Column 30 m × 0.25 mm × 0.25 μm) installed in a GC (7890B, Agilent) coupled with a time-of-flight MS (Pegasus® BT, LECO)]. The system was controlled by ChromaTOF® (LECO). Raw data was converted to netCDF format using Chemstation (Agilent), before the data was imported and processed in Matlab R2021b (Mathworks, Inc.) using the PARADISe[56] software. The metabolites were normalized by median of the data followed by a logarithmic transformation and scaled by auto (mean-centered and divided by the standard deviation of each variable) using MetaboAnalyst[57] v5.0. T-tests were performed with an FDR cutoff of 0.05.

## Statistics and reproducibility

No statistical methods were employed to predetermine sample sizes for the fish feeding trials or for the collection of salmon digesta samples. The experiments were not randomized. Statistical analyses were conducted in R (v4.2.2) and details of the experiments are indicated in each figure legend. For alpha and beta diversity analyses, the Shannon diversity index was employed to determine alpha diversity, while Bray–Curtis, weighted, and unweighted UniFrac metrics were used to determine beta diversity. Additional comparisons of community richness and diversity between groups were assessed using the Kruskal–Wallis test, while pairwise comparisons were performed with the Wilcoxon test, with Benjamini–Hochberg FDR multiple test correction applied. For comparisons involving more than two groups, the Kruskal–Wallis test was utilized, while the Wilcoxon test was applied for pairwise comparisons between two groups. *P*-values were adjusted for multiple testing using the FDR control. Differences were considered statistically significant when the *P* value was less than 0.05 (*$P < 0.05$), with additional thresholds set at **$P < 0.01$, ***$P < 0.001$, and ****$P < 0.0001$. No data were excluded from the analysis, and all available samples were included in the respective statistical assessments. The investigators were not blinded to the allocation during the experiments or the outcome assessment.

## Reporting summary

Further information on research design is available in the Nature Portfolio Reporting Summary linked to this article.

## Data availability

The raw metagenomics datasets for the low and high dose mannan trials are available in the Sequence Read Archive (SRA) repository under project ID PRJNA947090. The raw host transcriptomics and metatranscriptomics data for the low dose mannan trial are available under project IDs PRJEB73366

and PRJEB67787, respectively, and, for the high dose mannan trial under project IDs PRJNA1051365 and PRJNA1051380, respectively. The data used to generate Fig. 1 can be found in Supplementary Data 1and Supplementary Data 3. The data used to generate Fig. 2 can be found in Supplementary Data 2. The data used to generate Fig. 3 can be found in supplementary Supplementary Data 4d.

## Code availability
The code generated in this study is available at Github (https://github.com/shashank-KU/ImprovaFish-MDF-Effects) or Zenodo (https://doi.org/10.5281/zenodo.13919575).

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

## Acknowledgements

This work was supported by the Research Council of Norway (project no. 300846), the Swedish Research Council Formas (grant no. 2019-02336) and the European Union's Horizon 2020 research and innovation program under the ERA-Net Cofund project BlueBio (grant agreement no. 311913). The Danish National, Research Foundation grant no. DNRF143 to M.T.L. The Orion High Performance Computing Center at the Norwegian University of Life Sciences and Sigma2 - the National Infrastructure for High Performance Computing and Data Storage in Norway are acknowledged for providing computational resources that have contributed to meta-omics analyses described in this study. Jacob A. Rasmussen is thanked for help handling and depositing raw sequence data. The authors thank Elixir-Norway (NFR project no. 322392) for bioinformatics and data management related services.

## Author contributions

P.B.P., S.L.L.R., S.R.S., T.R.H. and S. B. designed the study. Transcriptomics and meta-transcriptomics data generation were done by M.T.L, C.G.C., L.P., and A.R.A.V. Transcriptomics and meta-transcriptomics analysis were done by S.G., A.V.P.-L. and M.K. 16S rRNA gene sequencing analysis were carried out by S.G. Metabolomic analyses were carried out by A.N., S.G., and S.L.L.R. T.N.H., and M.H. helped in the sequence analysis. Feeds and feed supplements were formulated by B. S., V.C., R.K., S.S. and B.W. Fish phenotypes were assessed by T.M. The draft manuscript was written by S.G., S.L.L.R., P.B.P, S.R.S., and T.R.H. All authors contributed to the editing of the text and content and approved the final version.

## Competing interests

B.S., V.C., R.K. and S.S. are employed at Cargill Group, who produces, markets and sells fish feed supplements with some of the ingredients tested in the current investigation. Furthermore, Cargill provided parts of the funding for the fish trials. S.L.L.R. is an Editorial Board Member for Communications Biology, but was not involved in the editorial review of, nor the decision to publish this article. All other authors declare no competing interests.
