## [Transparent Peer Review file · Communications Biology]

The need for high-resolution gut microbiome characterization to design efficient strategies for sustainable aquaculture production

Corresponding Author: Professor Sabina La Rosa

Version 0:

Reviewer comments:

Reviewer #1

(Remarks to the Author)

The paper provides a broad overview of Atlantic salmon omics, in order to characterize changes in the gut microbiome when fed with prebiotic additives. In the study, the authors performed a very large number of analysis, ranging from metataxonomic analysis to metabolomics, transcriptomics and host phenotypic outputs, which is quite impressive.

Despite the lack of significant overall results, the authors have made a good effort to provide some novel insights on the functionality of selected microbes within the fish gut. Overall the paper reads well and straightforward. I only have some clarification comments/suggestions.

Line 32: The authors describe the salmon microbiome as poorly characterized. I can see some truth in it, especially with regard to functionalities. But it is also a fact that many paper so far has been published using mainly 16S analysis, to describe the salmon microbiota. So the term as "poorly characterized" I am not sure it is entirely true. I would advice to specify this a bit more.

Figure 1G and 2G - the legend of the heatmap is incomplete. Please indicate what does the colour of the heatmap represent and what is the colour range from 3 to -3 or from 2 to -2 indicate. I assume is the Z score, but please indicate that in the legend

Overall in the PCoA in Figure 2 would be helpful to add a p value to show significance or not.

Fig 3: Maybe something I did not get right. *Limnosibacillus* and *Lactobacillus* were actually not really enriched with MN3 addition. So how exactly should one use this info on their functionality to shape with MDFs? Also how were the other microbes in the figure selected. This info I missed.

The legend in Fig 3 needs to be checked - some sentences read unfinished.

Reviewer #2

(Remarks to the Author)

The manuscript submitted by Gupta et al, describes an investigation into the effects of dietary MDFs on Atlantic salmon gut microbiome diversity, structure and function using a combination of 16S amplicon sequencing, metatranscriptomics and host transcriptomics. The authors report limited effects of these functional feeds on any of the parameters tested and focus on making recommendations for future work.

Firstly, I found the structure of the manuscript to be incomplete. There is no methods section and the results and discussion, combined, is very descriptive. It is difficult to ascertain what the key findings are.

Overall, I certainly see the value in performing the study and I agree that negative results are important to know about, however I do not think this manuscript presents any novel insights of value to the research field.

I do agree with some of the points made in the discussion, especially that differences observed using 16S rRNA amplicon sequencing are not necessarily translated into functional differences and that care should be taken into over-interpreting 16S results. But this point has been made elsewhere and with a more convincing discussion about functional redundancy etc.

Finally, I do not think that this manuscript gives a good representation of previous research that has been done on salmon microbiomes. The introduction (L74-86) says there is a striking lack of genomic information for the salmon gut microbiome, but does not acknowledge what is known about salmon microbiomes. They also imply they are the first to investigate the use of mannans as functional feeds in salmon, which is not true.

Version 1:

Reviewer comments:

Reviewer #1

(Remarks to the Author)

The authors have addressed most of my comments which improved the clarity of the manuscript. I have still some minor revisions that still need to be addressed:

Figure 1- Include also p values similar to Figure 2

Regarding the justification for figure 3, for the inclusion of *Limnosibacillus* and *Lactobacillus*, please add the information you provided to me and also make it clear in the text.

Also for figure 3, a clarification: you mention you selected 9 random taxa to include. But if those taxa represent high abundant and less abundant taxa, is this really random selection? Please make this clear.

Reviewer #3

(Remarks to the Author)

In their paper, Gupta et al describe the influence of MDFs on the Salmon microbiome using a variety of omics datasets. The authors did a good job visualizing and integrating these complex datasets. In addition, the article is nicely written and condenses the key results quite well.

I agree with Reviewer 2 that this study is very descriptive and does not present much novelty. However, omics datasets are very descriptive in nature and the authors did well in not over-interpreting their results. I see the value in the publication of this study as this will provide a rich dataset for future reference and a description of the (negative) results of mannans on the Salmon microbiome.

I have a few comments:

- Since Figure 3 is one of the main figures and selling points of the article, I would recommend to describe in the method section and/or legend in more detail how this figure was generated: How was the data mapped to CAZy? Did the authors just use the data that was mapped to SMGA MAGs or also the de-novo assembly? What is the difference between the red, brown, blue, and black arrows?

- Was transcriptomic data of the host used to confirm the potential uptake of SCFAs with actively transcribed transport proteins?

- The authors could discuss additional layers of complexity in fish farming that could also influence the microbiome: such as the difference between RAS and open net-pens, use of antibiotics, etc

Minor comments:

- L277: I would replace "in-depth mechanistic understanding" with "in-depth descriptive understanding" as the authors failed to demonstrate any causal mechanisms in their study

- L265-266: "The main carbohydrates predicted to be utilized (beta-glucans, xylans, galactans, and chitin) predicted to be utilized". Please delete one of the "predicted to be utilized"

- L30: "a priori" should be in cursive font as latin "ad libitum" in L344 is cursive as well

- Supplementary Figure 1/2/3 labels: "control/reference (CTR)" instead of "control/refence (CTR)". I would recommend describing this sample as "control" to be consistent with the rest of the manuscript

We thank the editor and the reviewers for their time and insightful comments. We feel that the changes made in light of their concerns have helped clarify and improve several aspects of the manuscript as well as showcase the novel insights of value to the research field.

We now submit a revised version of COMMSBIO-24-1294 that addresses and accommodates all of the comments. The review has been broken down and our point-by-point answers to the individual comments are dealt with chronologically. Our responses are in blue lettering. The line numbers refer to the “clean” version of the revised manuscript; however, we have also provided a version of the manuscript where the changes we have made have been tracked.

Reviewer #1

The paper provides a broad overview of Atlantic salmon omics, in order to characterize changes in the gut microbiome when fed with prebiotic additives. In the study, the authors performed a very large number of analysis, ranging from metataxonomic analysis to metabolomics, transcriptomics and host phenotypic outputs, which is quite impressive.

Despite the lack of significant overall results, the authors have made a good effort to provide some novel insights on the functionality of selected microbes within the fish gut. Overall, the paper reads well and straightforward. I only have some clarification comments/suggestions.

Re: We sincerely thank the reviewer for the positive remarks, their recognition of the merits of our study, and overall interest in the data presented in our manuscript. We very much appreciate the constructive comments and detailed suggestions for improving the manuscript.

Specific comments:

1) Line 32: The authors describe the salmon microbiome as poorly characterized. I can see some truth in it, especially with regard to functionalities. But it is also a fact that many paper so far has been published using mainly 16S analysis, to describe the salmon microbiota. So, the term as "poorly characterized" I am not sure it is entirely true. I would advise to specify this a bit more.

Re: We thank the reviewer for pointing out how our wording in the abstract may have been misleading. To avoid any potential misunderstanding, we have now amended the corresponding sentence to remove “poorly characterized” and instead provide a more precise description of the scope of our study. The revised text in the abstract (**Lines 29-32**) reads:

*“Here we used initial metataxonomic data on fish gut microbiomes as well as a wealth of a priori mammalian microbiome knowledge on α -mannooligosaccharides (MOS) and β -mannan-derived MDFs to study effects of such feed supplements in Atlantic salmon (*Salmo salar*) and their impact on its gut microbiome composition and functionalities.”*

Additionally, also in light of reviewer 2’s comments, we have expanded the introduction section to report the knowledge on the composition and potential functions of the salmon gut microbiome at the time this study began (see our response to comment #4 by reviewer 2, below).

2) Figure 1G and 2G - the legend of the heatmap is incomplete. Please indicate what does the colour of the heatmap represent and what is the colour range from 3 to -3 or from 2 to -2 indicate. I assume is the Z score, but please indicate that in the legend Overall in the PCoA in Figure 2 would be helpful to add a p value to show significance or not.

Re: We thank the reviewer for catching this. We have revised both **Figure 1G and 2G** as advised and added p-values in **Figure 2D-2E-2G-2H**.

The legend for the new version of **Figure 1**:

has been updated as follows (**Lines 157-180**):

“Fig. 1: Effect of mannan (0.2% inclusion rate) on host and gut microbial community structure and function. (A) Sampling strategy for studying the effect of mannan on the temporal dynamics of the Atlantic salmon gut microbiota. T0 (parr), T1 (pre-smolts), T2 (smolts), and T3 (post-smolts) represent the different sampling time points. The experimental groups are labeled as CTR (Control), MC1 (Diet 1), MC2 (Diet 2), and MN3 (Diet 3), indicating the different diets administered during the study. (B) The mean body weight in all the experimental groups, stratified by sampling time. (C) Alpha diversity (Shannon diversity index), stratified by sampling time. Boxplots show medians and IQR. P-values were determined by the Wilcoxon test with FDR control for multiple testing. (D) Beta diversity was assessed using Bray-Curtis dissimilarity for 16S rRNA gene data obtained from the hindgut samples. The effects of sampling time were tested with PERMANOVA. Each dot represents individual samples colored by sampling time (T0 (parr), T1 (pre-smolts), T2 (smolts), and T3 (post-smolts)), as indicated in the legend. (E) Beta diversity was assessed through Bray-Curtis dissimilarity for 16S rRNA gene data obtained from the hindgut samples. The effects of different diets were tested with PERMANOVA. Each dot represents individual samples colored by different MDFs (CTR (Control), MC1 (Diet 1), MC2 (Diet 2), and MN3 (Diet 3)), as indicated in the legend. (F) PCA plot showing the differences in host gene

expression between MDFs (MC1 (Diet 1), MC2 (Diet 2), and MN3 (Diet 3)) and control samples (CTR) from the hindgut. The numbers on the axes represent the variance explained by the principal components. (G) Top 20 most abundant genera in all the groups based on 16S rRNA gene data. Other genera with relative abundance less than 1% were shown as “Other genus”. Statistically significant differences between MDFs (MC1 (Diet 1), MC2 (Diet 2), and MN3 (Diet 3)) and control samples (CTR) were calculated using the Wilcoxon test, with significance levels indicated by stars: * $p \leq 0.05$, ** $p \leq 0.01$, *** $p \leq 0.001$. Colors represent row z-scores of each microbial genus (brown: high; turquoise: low). (H) PCA plot showing the differences in metatranscriptome expression between MDFs (MC1 (Diet 1), MC2 (Diet 2), and MN3 (Diet 3)) and control samples (CTR) from the hindgut. The numbers on the axes represent the variance explained by the principal components.”

The legend for the revised version of **Figure 2**:

has been updated and reads (Lines 218-239):

“Fig. 2: Effect of β -mannan (4% inclusion rate) on host and gut microbial community structure and function. (A) Sampling strategy for studying the effect of β -mannan on the temporal dynamics of the Atlantic salmon gut microbiota. The experimental groups are labeled as CTR (Control) and 4%MN3 diet (4% β -mannan diet). (B) Boxplot showing the mean body weight

in all the experimental groups. Boxplots demonstrate medians and IQR. P-values were determined by the Wilcoxon test with FDR control for multiple testing. (C) Alpha diversity (Shannon diversity index), stratified by sample type (hindgut and pyloric caeca). Boxplots demonstrate medians and IQR. P-values were determined by the Wilcoxon test with FDR control for multiple testing. (D) Beta diversity was assessed using Bray-Curtis dissimilarity for 16S rRNA gene data obtained from the hindgut samples. The effects of the diet were tested with PERMANOVA. Each dot represents individual samples colored by diet, as detailed in the legend. (E) Beta diversity was assessed using Bray-Curtis dissimilarity for 16S rRNA gene data obtained from the pyloric caeca samples. The effects of the diet were tested with PERMANOVA. Each dot represents individual samples colored by diet, as detailed in the legend. (F) Top 20 most abundant genera in all the groups based on 16S rRNA gene data. Other genera with relative abundance less than 1% were shown as "Other genus". Statistically significant differences between the 4% β -mannan (4%MN3) and control (CTR) samples were calculated using the Wilcoxon test, with significance levels indicated by stars: * $p \leq 0.05$, ** $p \leq 0.01$, *** $p \leq 0.001$. Colors represent row z-scores of each microbial genus (brown: high; turquoise: low). (G) PCA plot showing the differences in gene expression between the 4% β -mannan (4%MN3) and control samples from the hindgut. The numbers on the axes represent the variance explained by the principal components. (H) PCA plot showing differences in metatranscriptome expression between 4% β -mannan (4%MN3) and control (CTR) samples from the hindgut. The numbers on the axes represent the variance explained by the principal components."

3) Fig 3: Maybe something I did not get right. *Limnosibacillus* and *Lactobacillus* were actually not really enriched with MN3 addition. So how exactly should one use this info on their functionality to shape with MDFs?

Re: The reviewer is correct in pointing out that *Limosilactobacillus* and *Lactobacillus* were not enriched when MN3 (or MC1/MC2) was supplemented in the salmon diet. Our intent in **Figure 3** (legend at **Lines 269-279**) was to provide a comprehensive description of the results obtained using a genome-resolved metatranscriptomic approach that has allowed to identify active pathways enabling microbial catabolism of complex glycans, and not limited to microbial processing of acetylated-galactoglucomannan (MN3) or alpha-mannans. Indeed, the acetylated-galactoglucomannan (MN3) did not elicit upregulation of genes encoding enzymes with functions compatible with utilization of this carbohydrate by the salmon gut microbiota, as stated at **Lines 281-284**, and for this reason the mannan structures are not reported in **Figure 3**, nor enzymes or pathways related to its degradation are displayed in the representations of *Limosilactobacillus* and *Lactobacillus*. Instead, based on the re-analysis of our integrated multi-omic dataset, we have chosen to present data for selected metabolically active microbes reported to be important resident of the salmon gut in previous studies, including *Limosilactobacillus* and *Lactobacillus* that have also been explored for the possibility of being used as probiotics to enhance growth performance and feed efficiency in Atlantic salmon **[Cathers 2022]**. **Figure 3** shows that *Lactobacillus* and *Limosilactobacillus* could actively metabolize xyloglucan and galactans, as genes encoding xylosidases, glucosidases, and galactosidases were found to be present in their genome and being upregulated. As such, given that these enzymes act on specific residues, linkages and positions on the carbohydrate chains, MDFs based on a backbone of i) b-1,4-linked glucose and decorations that include galactose, xylose and arabinose, or ii) b-1,3-linked or b-1,4-

linked galactose, could be developed to match the enzymatic toolbox of these lactic-acid producing microbes, eventually promoting their growth and beneficial outputs.

4) Also how were the other microbes in the figure selected. This info I missed.

Re: The reviewer is referring to how the microbes represented in **Figure 3** were selected. The microbes belong to the genera *Lactobacillus*, *Limosilactobacillus*, *Photobacterium*, *Pseudomonas*, *Serratia*, *Glutamicibacter*, *Cetobacterium*, *Paraburkholderia* and *Sphingomonas*. These organisms were chosen to represent common and diverse metabolisms and phylogenetic units known to be present in salmon gut samples. Indeed, publicly available studies based on 16S rRNA amplicon sequencing have described members of the order Lactobacillales (i.e. *Lactobacillus*, *Limosilactobacillus*, etc.), Enterobacterales (i.e. *Photobacterium* and *Alivibrio*) and Pseudomonadota (i.e. *Pseudomonas* and *Burkholderia/Paraburkholderia*), as the more prevalent bacterial genera in the salmon gut [Bozzi 2021]; [Weththasinghe 2022]; [Wang 2021]. Additionally, less abundant genera detected in the salmon gut microbiome include *Cetobacterium*, *Sphingomonas*, *Shewanella*, *Serratia*, *Lelliota* and Actinomycetales (*Glutamicibacter* spp) [Li 2022]; [Weththasinghe 2022]; [Wang 2018]; [Wang 2021]; [Fogarty 2019]; [Kazlauskaite 2021]. The genomes of these organisms are included in our Salmon gut Microbial Genome Atlas [De León 2023], that has allowed to generate the genome-resolved metatranscriptomic dataset presented in this study.

To create the portrayal of the metabolic features of these salmon gut microbes from the chosen genera, we randomly selected a subset of representatives of these taxa that were detected as metabolically active in the genome-resolved metatranscriptomic data. For this subset, we manually parsed carbohydrate active enzymes, and based on the predicted functions, substrate utilization and downstream metabolism of individual monosaccharides was inferred, while calling and confirming that each gene involved in individual metabolic pathways was detected as differentially expressed in the metatranscriptomics data.

We have clarified this more explicitly in the text at Lines 247-251:

“From the 211 microbes detected as metabolically active in the metatranscriptomics datasets (Supplementary Fig. S6 and S10), we used a subset of 9 randomly selected bacteria that represent high abundant and less abundant genera in the salmon gut microbes, as described in publicly available studies [Bozzi 2021]; [Weththasinghe 2022]; [Wang 2021]; [Weththasinghe 2022]; [Wang 2018]; [Wang 2021]; [Fogarty 2019]; [Kazlauskaite 2021].”

5) The legend in Fig 3 needs to be checked - some sentences read unfinished.

Re: As per reviewer’s comment, the legend in **Figure 3** has been amended to improve readability. The revised text at Lines 269-279 now reads:

“Fig. 3. Selected metabolic features of the salmon gut microbiome of adult fish as inferred from genome and metatranscriptome comparisons. The different metabolic pathways, including host and dietary carbohydrate depolymerization, glycolysis, tricarboxylic acid (TCA) cycle and SCFA production, are displayed for each population MAG. The graphical representation includes different carbohydrates, CAZymes, and cellular features based on functional annotations are depicted as numbered boxes or abbreviated gene names, which are additionally listed in

Table S1d. Features are included if a gene was expressed at either the smolt (T2) or adult (T3) stage from either the control or MDF (MC1, MC2, MN3) diets. The main carbohydrates predicted to be utilized (beta-glucans, xylans, galactans, and chitin) predicted to be utilized, SCFAs (e.g., acetate), and organic acids (e.g., lactate and succinate) are represented by large colored arrows. GTBD-Tk inferred taxonomy is included. Gene names and abbreviations are also provided in **Table S1d.**”

Reviewer #2

The manuscript submitted by Gupta et al, describes an investigation into the effects of dietary MDFs on Atlantic salmon gut microbiome diversity, structure and function using a combination of 16S amplicon sequencing, metatranscriptomics and host transcriptomics. The authors report limited effects of these functional feeds on any of the parameters tested and focus on making recommendations for future work.

Re: We thank the reviewer for the careful reading, overall interest in data, and suggestions to further improve the manuscript. We recognize the reviewer’s concerns which we have responded to in full, and in the process have significantly strengthened the original version of the study.

Specific comments:

1) Firstly, I found the structure of the manuscript to be incomplete. There is no methods section and the results and discussion, combined, is very descriptive. It is difficult to ascertain what the key findings are.

Re: We have now moved the Methods section from the Supplementary Materials to the main text. The manuscript, consistent with others in Communications Biology, is written as combined results and discussion. We hope this reviewer agrees that this is a rich dataset that can be presented in many different ways. A detailed description of all the different omic layers in the main text would be overwhelming for the reader, and for this reason we choose to keep it as Supplementary Results. However, to account for this reviewer’s insight, we have now included and expanded a **Conclusion** section to improve clarity of the key findings of our study (**Lines 281-315**):

“Although we observed a scarcity of mannan-degrading capabilities, this study achieved for the first time an in-depth mechanistic understanding for how diet and specific microbial populations interact, which should inform future MDF design and testing of their effects on the salmon holobiont. Notably, while metataxonomy data indicated an effect of the mannan diet on the Burkholderia-Caballeronia-Paraburkholderia population, these results did not go beyond bacterial proportions and no empirical evidence of diet-microbiome interaction was obtained. Instead, our study revealed the active roles of beneficial microbial groups, such Lactobacillus and Limosilactobacillus, highlighting their contributions to nutrient utilization, potentially producing metabolites associated with host health states [Cathers 2022]. We hypothesize that trials with MDFs such as pectin-derived galactans or (arabino)xyloligosaccharides designed around this new microbiome information will result in more concrete insights about the metabolic roles of

beneficial gut associated microbiota and host effects in salmon and other fish in the instances they are implemented.”

2) Overall, I certainly see the value in performing the study and I agree that negative results are important to know about, however I do not think this manuscript presents any novel insights of value to the research field.

Re: We are grateful to the reviewer for recognizing the importance of the presented study, and we appreciate the opportunity to add further clarity on the novelty of our research that would be of interest to the scientific community. First of all, we would like to point out that while the essential role of gut microbiomes has been repeatedly showcased for health, nutrition and productivity in humans and terrestrial livestock, the knowledge on how gut microbes support fish such as Atlantic salmon remains sparse. Current knowledge on salmon gut microbiota is primarily based on 16S rRNA gene amplicon surveys (from *in vitro* and *in vivo* experiments) that do not allow to obtain insight into the real functions of the endogenous gut microbiome during fish growth and when exposed to different feed regimens. While meta-taxonomy only allows to make functional predictions and hypothesis (that are of limited importance if not verified), our study offers detailed mechanistic insights into dietary fiber utilization of the salmon endogenous gut microbiome and its resulting metabolic output.

Specifically:

- we present a high resolution multi-omic approach demonstrating that α -mannan and β -mannan, supplemented at varying doses, have negligible effects on both host gene expression and the structure and function of the salmon gut microbiome. We conclude that these specific MDFs do not qualify for further research as novel feed ingredients aimed at stimulating microbes present under normal fish rearing conditions;

- we demonstrate the value of direct evidence provided by the application of our genome Salmon Gut Microbial Genome Atlas [De León 2023] that finally enable functional studies of the salmon endogenous gut microbiome. We showcase a model approach outlining the blueprint for future tests aimed at devising effective strategies to select feed additives holding the potential of microbiome-reprogramming outcomes in the salmon gut;

- we reconstruct, for the first time in the field, metabolic dynamics at microbial community level in the salmon gut and used this knowledge to predict and outline novel and potentially beneficial endogenous lactic acid bacteria that should be targeted with future, conceivably more suitable, MDF strategies for salmon.

As such, we believe our results will be of significant interest to the research field focused on salmon aquaculture and to a broad audience, including scientists working within the fields of microbiome, holo-omics, animal nutritional researchers.

3) I do agree with some of the points made in the discussion, especially that differences observed using 16S rRNA amplicon sequencing are not necessarily translated into functional differences and that care should be taken into over-interpreting 16S results. But this point has been made elsewhere and with a more convincing discussion about functional redundancy etc.

Re: We do not claim to be the first to report the finding that 16S rRNA amplicon sequencing gives limited insight and detected differences in relative abundance do not always translate into difference in functions. There are several previous studies about this limitation, primarily discussed in other gut ecosystems, as well as recent studies in salmon, which we cite [Li 2022]; [Karlsen 2022], that highlights that the composition of the gut microbiota closely resembled that of the feed. Despite this, the salmon gut microbiome field continues to be completely dominated by metataxonomy surveys that do not discriminate between metabolically active and inactive populations in the gut (including carry-over of microbial DNA from abundant microbial populations in the feed), and result interpretation is based on predicted metabolic functions obtained from matching bacteria in non-salmon gut ecosystems (primarily on knowledge obtained from human gut bacteria).

The approach presented in our study clearly uncovers unaffected and metabolic activated microbial populations, as well as to decrypt active microbial mechanisms for carbohydrate degradation that could only be detected due to the availability of our salmon gut-specific microbial genome collection. Clearly, our systematic effort to disclose the functional content encoded within gut microbial genomes finally enables studies for understanding salmon host-microbiome ecology as well as to identify active microbial mechanisms that affect nutrient utilization and fish health status.

We posit that our study is the most comprehensive study to date, where we have used the SMGA [De León 2023], our unique genomic resource, to conduct, for the first time, genome-resolved metatranscriptomics, and studied how the microbiome composition and real functions vary across different life stages in salmon and when faced by different feed supplements. Despite the negative results, the integration of metatranscriptomics and host transcriptomics has explored possible interactions of bacterial outputs with the salmon, paving the way for further studies to mechanistically unveil how observed changes in microbiome composition and activities directly translate into functional alterations that impact holobiont metabolism.

That being said, the novelty of our work is not the observation that microbial compositional changes do not necessarily translate into functional differences, but the fact that is the first study to report functional meta-omic studies on the salmon gut ecosystem, as, to the best of our knowledge, no metatranscriptomic or metaproteomic work has been reported to date. Clearly, our effort to disclose the functional content encoded within gut microbial genomes is a game-changer in the field and will enable further studies for understanding salmon host-microbiome ecology as well as to identify active microbial mechanisms, and possibly novel biological functions, that affect nutrient utilization and fish health status.

We have revised the text in the **Conclusion** section (Lines 280-315) to further clarify the novelty aspects of our manuscript in light of this reviewer's comment.

4) Finally, I do not think that this manuscript gives a good representation of previous research that has been done on salmon microbiomes. The introduction (L74-86) says there is a striking lack of genomic information for the salmon gut microbiome, but does not acknowledge what is known about salmon microbiomes.

Re: As per reviewer suggestion, we have amended the text in the introduction to acknowledge what genomic information for the salmon gut microbiome was available prior to the present study. In the introduction, we have now written (at **Lines 74-100**):

*“Despite preliminary indications via taxonomic surveys that *Carnobacterium*, *Roseburia* and *Faecalibacterium* spp. exist in the salmon gut [Strand 2021], a striking paucity of genomic information for the salmon gut microbiome [Legrand 2020] has so far prevented the in-depth evaluation of the effectiveness of MDFs such as α - and β -mannans to match enzymatic abilities inherent to endogenous gut microbes. Indeed, at the inception of the present study, the understanding of the functional potential of the salmon gut microbiome was limited to microbes belonging to two salmon gut-associated genera, namely *Lactobacillus* and *Mycoplasma*, and derived from the genomes of 19 gut-associated *Lactobacillus* isolates [Cathers 2022] and 11 *Mycoplasma* metagenome assembled genome (MAGs) [Rasmussen 2021]; [Rasmussen 2023] that met the medium-quality level of the Minimum Information about a Metagenome Assembled Genome (MIMAG) criteria (completion $\geq 50\%$, contamination $\leq 5\%$) [Bowers 2017]. More recently, our group has established the first resource of metagenomes and genomes from cultured bacterial strains from the salmon gut, namely the Salmon gut Microbial Genome Atlas (SMGA), an assemblage of 211 closed bacterial genomes and medium/high-quality MAGs obtained through cultivation and shotgun metagenomics using long-read and short-read sequencing. Importantly, this resource captures the compositional and metabolic diversity of the salmon gut ecosystem, that, based on 16S rRNA amplicon sequencing, includes members of the order *Lactobacillales* (i.e *Lactobacillus*, *Limosilactobacillus*, etc.), *Enterobacterales* (i.e *Photobacterium* and *Alivibrio*) and *Pseudomonadota* (i.e *Pseudomonas* and *Burkholderia/Paraburkholderia*), as the more prevalent bacterial genera [Bozzi 2021]; [Weththasinghe 2022]; [Wang 2021], as well as less abundant genera such as *Cetobacterium*, *Sphingomonas*, *Shewanella*, *Serratia*, *Lelliota* and *Actinomycetales* (*Glutamicibacter* spp) [Wang 2021]; [Kazlauskaite 2021]; [Weththasinghe 2022]; [Fogarty 2019]; [Wang 2021],. While studies limited to the assessment of compositional changes in salmon gut studies have often been biased by carry-over of microbial DNA from abundant microbial populations in the feed [Karlsen 2022]; [Li 2022], application of the SMGA as a database for functional omics finally enables to clearly uncover dietary effects of feed interventions, discriminating between unaffected and metabolic activated microbial populations interacting with dietary components.”*

5) They also imply they are the first to investigate the use of mannans as functional feeds in salmon, which is not true.

Re: We thank the reviewer for bringing this point to our attention and for the opportunity to add clarity on this important aspect of the study. Since no reference has been mentioned, we did our best to infer the reviewer’s intent (please feel free to reiterate if we misaddressed this concern). In our salmon trials, **two types of mannans** were used: 1) alpha-mannan (MC1 and MC2) and 2) acetylated galactoglucomannan (MN3). Both **mannans are polymeric carbohydrates**, meaning their structures have a degree of polymerization higher than 10 and up to 150 sugar monomers.

Re. the previous use of alpha-mannan as prebiotic for salmon: to the best of our knowledge, only alpha-MannoOligosaccharideS (MOS; also known with the commercial name of Bio-MOS)

have been tested as functional feeds for salmon, both *in vitro* [Kazlauskaite 2022] and *in vivo* (Dimitroglou 2011); [Leclercq 2020]; [Grisdale-Helland 2008). Structurally, an alpha-MOS consists of less than 10 sugar monomers.

Re. the previous use of acetylated galactoglucomannan as prebiotic for salmon: as far as we are aware, no study has tested acetylated galactoglucomannan as functional feed for salmon. Please note that this complex carbohydrate is unique, as it combines the structural features of commercially available beta-mannans (linear mannans, glucomannans and galactomannans) and it is not available on the market. Indeed, this complex carbohydrate is produced *in-house*, in our biorefinery facility at the Norwegian University of Life Sciences, from spruce wood, as described previously [La Rosa 2019]. Briefly, the process pretreats wood sawdust by steam explosion, followed by ultrafiltration and nanofiltration to enrich for polymeric carbohydrates in the mixture. Structurally the acetylated galactoglucomannan is totally different from the MOS used in published work, the latter consisting of a short carbohydrate chain (2-10 sugar monomers) of alpha-1,6/alpha-1,3/alpha-1,2 linked mannose residues. Our acetylated galactoglucomannan consists of a backbone with beta-1,4 linked glucose and mannose residues, with alpha-1,6 linked galactose substitutions and is acetylated at positions O-2, O-3, and/or O-6. Degree of polymerization of the acetylated galactoglucomannan is around 150 sugar monomers.

As such, we hope this reviewer agrees that the potential effects on the salmon host and the enzymatic toolbox deployed by the gut microbiota to interact with the functional feeds used in our study are completely different from those elicited by the alpha-mannooligosaccharides (MOS) used in previously published studies, and our investigations are novel.

If we are missing other published studies, we would be happy to obtain the references and will make sure to include them.

Bibliography

[Cathers 2022] Cathers HS, et al. *In silico, in vitro* and *in vivo* characterization of host-associated *Latilactobacillus curvatus* strains for potential probiotic applications in farmed Atlantic salmon (*Salmo salar*). Sci Rep. 2022 Nov 1;12(1):18417.

[Bozzi 2021] Bozzi D, et al. Salmon gut microbiota correlates with disease infection status: potential for monitoring health in farmed animals. Anim Microbiome. 2021 Apr 20;3(1):30.

[Weththasinghe 2022] Weththasinghe P, et al. Modulation of Atlantic salmon (*Salmo salar*) gut microbiota composition and predicted metabolic capacity by feeding diets with processed black soldier fly (*Hermetia illucens*) larvae meals and fractions. Anim Microbiome. 2022 Jan 15;4(1):9.

[Wang 2021] Wang J, et al. Microbiota in intestinal digesta of Atlantic salmon (*Salmo salar*), observed from late freshwater stage until one year in seawater, and effects of functional ingredients: a case study from a commercial sized research site in the Arctic region. Anim Microbiome. 2021 Jan 28;3(1):14.

[Li 2022] Li Y, et al. Consistent changes in the intestinal microbiota of Atlantic salmon fed insect meal diets. Anim Microbiome. 2022 Jan 10;4(1):8.

- [Wang 2018]** Wang C., et al. Intestinal microbiota of healthy and unhealthy Atlantic salmon *Salmo salar* L. in a recirculating aquaculture system. *J. Ocean. Limnol.* 36, 414–426 (2018).
- [Fogarty 2019]** Fogarty C, et al. Diversity and composition of the gut microbiota of Atlantic salmon (*Salmo salar*) farmed in Irish waters. *J Appl Microbiol.* 2019 Sep;127(3):648-657.
- [De León 2023]** De León, AVP, et al. The Salmon Microbial Genome Atlas enables novel insights into bacteria-host interactions via functional mapping. *bioRxiv* 2023.12.10.570985.
- [Strand 2021]** Strand MA *et al.* Transkingdom network analysis provides insight into host-microbiome interactions in Atlantic salmon. *Comput Struct Biotechnol J* 19, 1028-1034 (2021).
- [Legrand 2020]** Legrand TPRA, et al. A microbial sea of possibilities: current knowledge and prospects for an improved understanding of the fish microbiome. *Rev Aquacult* 12, 1101-1134 (2020).
- [Rasmussen 2023]** Rasmussen JA et al. Co-diversification of an intestinal Mycoplasma and its salmonid host. *ISME JI* (2023). <https://doi.org/10.1038/s41396-023-01379-z>.
- [Rasmussen 2021]** Rasmussen JA et al. Genome-resolved metagenomics suggests a mutualistic relationship between Mycoplasma and salmonid hosts. *Comms Biol* 4, 579 (2021).
- [Bowers 2017]** Bowers RM et al. Minimum information about a single amplified genome (MISAG) and a metagenome-assembled genome (MIMAG) of bacteria and archaea. *Nat Biotech* 35, 725-731 (2017).
- [Kazlauskaite 2021]** Kazlauskaite, R. et al. SalmoSim: the development of a three-compartment in vitro simulator of the Atlantic salmon GI tract and associated microbial communities. *Microbiome* 9, 179 (2021).
- [Karlsen 2022]** Karlsen C, et al. Feed microbiome: confounding factor affecting fish gut microbiome studies. *ISME Commun.* 2022 Feb 2;2(1):14.
- [Kazlauskaite 2022]** Kazlauskaite R, et al. Deploying an *in vitro* gut model to assay the impact of the mannan-oligosaccharide prebiotic bio-mos on the Atlantic Salmon (*Salmo salar*) gut microbiome. *Microbiol Spectr.* 2022 Jun 29;10(3):e0195321.
- [Dimitroglou 2011]** Dimitroglou A, et al. The effect of mannan oligosaccharide supplementation on Atlantic Salmon smolts (*Salmo salar* L.) fed diets with high levels of plant proteins. *J Aquac Res Development* 2011, S1:011.
- [Leclercq 2020]** Leclercq E, et al. Dietary supplementation with a specific mannan-rich yeast parietal fraction enhances the gut and skin mucosal barriers of Atlantic salmon (*Salmo salar*) and reduces its susceptibility to sea lice (*Lepeophtheirus salmonis*). *Aquaculture* 529 (2020) 735701.
- [Grisdale-Helland 2008]** Grisdale-Helland B, et al. The effects of dietary supplementation with mannanoligosaccharide, fructooligosaccharide or galactooligosaccharide on the growth and feed utilization of Atlantic salmon (*Salmo salar*). *Aquaculture* 283 (2008) 163-167.
- [La Rosa 2019]** La Rosa SL, et al. Wood-derived dietary fibers promote beneficial human gut microbiota. *mSphere.* 2019 Jan 23;4(1):e00554-18. doi: 10.1128/mSphere.00554-18.

We wish to express our sincere appreciation to the editor and the reviewers for their comments and suggestions on our revised manuscript. We now submit a revised version of COMMSBIO-24-1294A that addresses and accommodates the concerns raised by referee #1 and #3. The review has been broken down and our point-by-point answers to the individual comments are dealt with chronologically. Our responses are in blue lettering. The line numbers refer to the “clean” version of the revised manuscript. In addition, we have provided a version of the manuscript where the changes we have made have been tracked.

Reviewer #1

The authors have addressed most of my comments which improved the clarity of the manuscript.

Re: We thank the reviewer for their valuable comments and positive evaluation of our revised manuscript.

I have still some minor revisions that still need to be addressed:

1) Figure 1- Include also p values similar to Figure 2.

Re: Done as suggested.

2) Regarding the justification for figure 3, for the inclusion of *Limnosibacillus* and *Lactobacillus*, please add the information you provided to me and also make it clear in the text.

Re: As the information provided to the reviewer were already included in the previous version of the manuscript at Lines 255-266, we have edited part of the text to further improve clarity and explicitly repeat the concept of a microbiota-directed fiber applied the results shown in Figure 3 (**Lines 225-231**):

“As such, this information could be extrapolated to develop sustainable LAB-specific MDFs, by matching the chemical structure of the feed supplement to the enzymatic toolbox of these lactic-acid producing microbes (that includes enzymes able to hydrolyze carbohydrates consisting of β -1,4-linked glucose and decorated with galactose, xylose and arabinose units, or polymers consisting of β -1,3-linked or β -1,4-linked galactose with arabinose substitutions), eventually promoting their growth and beneficial outputs³³”.

3) Also for figure 3, a clarification: you mention you selected 9 random taxa to include. But if those taxa represent high abundant and less abundant taxa, is this really random selection? Please make this clear.

Re: we have further clarified this aspect. Given that the high abundant and less abundant genera in the salmon gut each includes many species, we randomly selected bacterial species to represent the different relevant genera. The sentence at **Lines 198-202** now reads:

“From the 211 microbes detected as metabolically active in the metatranscriptomics datasets (Supplementary Figure S6 and S10), we used a subset of 9 randomly selected bacterial species from high abundant and less abundant genera in the salmon gut microbes, as described in publicly available studies^{4,20,21,23}.”

Reviewer #3

In their paper, Gupta et al describe the influence of MDFs on the Salmon microbiome using a variety of omics datasets. The authors did a good job visualizing and integrating these complex

datasets. In addition, the article is nicely written and condenses the key results quite well. I agree with Reviewer 2 that this study is very descriptive and does not present much novelty. However, omics datasets are very descriptive in nature and the authors did well in not over-interpreting their results. I see the value in the publication of this study as this will provide a rich dataset for future reference and a description of the (negative) results of mannans on the Salmon microbiome.

Re: We thank the reviewer for the careful reading, suggestions to improve the manuscript, overall positive attitudes toward our work and for recognizing the value of the rich dataset provided in this study for the fish gut microbiome community.

I have a few comments:

1) Since Figure 3 is one of the main figures and selling points of the article, I would recommend to describe in the method section and/or legend in more detail how this figure was generated: How was the data mapped to CAZy? Did the authors just use the data that was mapped to SMGA MAGs or also the de-novo assembly? What is the difference between the red, brown, blue, and black arrows?

Re: We thank the reviewer for this comment, and we have added relevant information to the method section “Metatranscriptomics: RNA extraction, library preparation, sequencing, and processing” based on the suggestions provided.

The new text at **Lines 491-500** reads as follows:

*“To create the portrayal of the metabolic features of 9 salmon gut microbes shown in Figure 3, we randomly selected a subset of bacterial species that represent high abundant and less abundant genera in the salmon gut microbes, and that were detected as metabolically active in the genome-resolved metatranscriptomic data. For this subset, we manually parsed carbohydrate active enzymes⁵⁵, and based on the predicted functions, substrate utilization and downstream metabolism of individual monosaccharides was inferred, while calling and manually confirming that each gene involved in individual metabolic pathways was detected as differentially expressed in the metatranscriptomics data. The complete list of gene names and abbreviations depicted in Figure 3 is provided in **Table S1d**.”*

Indeed, the data was mapped to the SMGA, that includes 68 MAGs that were generated using de-novo assembly from samples derived from this feeding trial [**De León 2023**]. This has now been clarified at **Lines 466-469** to be immediately available to the *Comms Biol* readership:

“These reads were used to quantify the expression of ORFs encoded by the genomes and MAGs included in the Salmon Microbiome Genome Atlas (SMGA; that also includes 68 MAGs obtained from samples derived from the low dose mannan trial)¹⁹ using kallisto⁵¹ (v0.44.0).”

The red, brown, blue, and black arrows are matched to different short chain fatty acids and organic acids, but there is no difference between them as they all indicate secretion of different compounds.

2) Was transcriptomic data of the host used to confirm the potential uptake of SCFAs with actively transcribed transport proteins?

Re: We thank the reviewer for the great suggestion. We have examined the salmon transcriptomic results and identified active transcription of genes encoding transport proteins involved in the uptake of short-chain fatty acids (SCFAs) and organic acids. To account for this comment, the following text has been added at **Lines 207-214**:

“Moreover, salmon RNA sequencing data revealed active transcription of transporters that could facilitate the uptake of microbial fermentation products (Supplementary Data 3). This included the monocarboxylate transporter 1 (MCT1) SLC16A1 [Sivaprakasam 2017] that facilitate uptake of lactate, propionate, and acetate into colonic cells. Similarly, transporters belonging to the SLC13 family, that mediates succinate uptake [Fremder 2021], were also expressed, collectively supporting the notion that the host may efficiently absorbs SCFAs and organic acids produced by microbial fermentation of dietary fibers.”

3) The authors could discuss additional layers of complexity in fish farming that could also influence the microbiome: such as the difference between RAS and open net-pens, use of antibiotics, etc.

Re: We appreciate the reviewer suggestion, but we opted to keep the introduction of our manuscript focused on the two main topics discussed: microbiota directed fibers and limited functional understanding of the salmon gut microbiota. Environmental factors, such as farm management practices and medical interventions are beyond the scope of this manuscript.

4) L277: I would replace "in-depth mechanistic understanding" with "in-depth descriptive understanding" as the authors failed to demonstrate any causal mechanisms in their study.

Re: We respectfully disagree with the reviewer, as we indeed failed to demonstrate any causal mechanisms related to beta-mannan utilization by salmon gut bacteria, but we report the in-depth mechanistic understanding, based on real in-vivo gene expression, of utilization of several other carbohydrates in the salmon gut microbiome. We therefore believe that the sentence at **Lines 239-242** is accurate:

“Although we observed a scarcity of mannan-degrading capabilities, this study achieved for the first time an in-depth mechanistic understanding for how diet and specific microbial populations interact, which should inform future MDF design and testing of their effects on the salmon holobiont”.

5) L265-266: "The main carbohydrates predicted to be utilized (beta-glucans, xylans, galactans, and chitin) predicted to be utilized". Please delete one of the "predicted to be utilized".

Re: We thank the reviewer for catching this. The sentence now reads:

“The main carbohydrates predicted to be utilized (beta-glucans, xylans, galactans, and chitin), SCFAs (e.g., acetate), and organic acids (e.g., lactate and succinate) are represented by large colored arrows”.

6) L30: "a priori" should be in cursive font as latin "ad libitum" in L344 is cursive as well.

Re: Done as suggested.

7) Supplementary Figure 1/2/3 labels: "control/reference (CTR)" instead of "control/refence (CTR)". I would recommend describing this sample as "control" to be consistent with the rest of the manuscript.

Re: Revised as per reviewer's suggestion.

Bibliography

[De León 2023] De León, AVP, et al. The Salmon Microbial Genome Atlas enables novel insights into bacteria-host interactions via functional mapping. bioRxiv 2023.12.10.570985

[Sivaprakasam 2017] Sivaprakasam S, et al. Short-Chain Fatty Acid Transporters: Role in Colonic Homeostasis. Compr Physiol. 2017 Dec 12;8(1):299-314.

[Fremder 2021] Fremder M, et al. A transepithelial pathway delivers succinate to macrophages, thus perpetuating their pro-inflammatory metabolic state. Cell Rep. 2021 Aug 10;36(6):109521.